# COMPUTE-EFFICIENT EVALUATION OF LLM VOTING ACCURACY

## ABSTRACT

Test-time scaling methods, such as voting, have emerged as a powerful paradigm to dramatically improve the performance of large language models (LLMs). Majority voting is often useful however to estimate the tradeoff between task performance (e.g., accuracy) and computational cost, as we vary the size of ensemble used in voting, denoted $M$; or as we vary hyperparameters, such as Temperature, in pursuit of a more favorable tradeoff. In the literature, evaluating voting accuracy performance is done using a purely empirical approach that requires many LLM evaluations and is highly computationally intensive. In this work we propose two methods to estimate the voting accuracy of an LLM with substantially less computational cost than current methods. Using a popular public benchmark datasets of LLM problems (MATH) we demonstrate that our two estimation approaches can closely approximate the true ensemble accuracy, with substantially less computational cost than current methods less computation than a purely empirical approach, especially as the number of votes grows larger.

## 1 INTRODUCTION

In recent years, large language models (LLM) have garnered tremendous attention due to their impressive capabilities on a wide variety of tasks, and their rapid improvement. Test-time scaling (Lewkowycz et al. (2022);Wang et al. (2023)) has emerged as a powerful paradigm to dramatically improve the performance of LLMs without further costly training by intelligently combining LLM inputs or outputs during inference (i.e., "test time"). One of the most widely-adopted test-time scaling methods is **test-time voting (TTV)**, wherein (typically) the same LLM is repeatedly given the same prompt so that it produces a *set* of $M$ solutions to that prompt, rather than a single solution. The final answer is then obtained by taking the most frequently-occurring solution (i.e., a plurality vote), or a solution that appears more than half of the time (i.e., a majority vote). The benefits of voting accuracy rely upon randomness in the output of the LLM, which is typically introduced by sampling output tokens from the posterior distribution that is predicted by the LLM. The degree of randomness in the output is also often mediated by a hyperparameter of the LLM (typically Temperature (Ackley et al. (1985);Bengio et al. (2000);Radford et al. (2019)) or Top_p/Nucleus Sampling (Holtzman et al. (2020)), which adjusts the entropy of the predicted posterior distribution.

Typically the performance benefits of TTV vary in proportion to $M$, and therefore there is a tradeoff between performance and computational cost. This tradeoff can also be made more, or less, favorable depending upon factors such as the difficulty of user prompts, and LLM hyperparameter settings (e.g., Temperature/Top_p). In practice therefore, given a representative sample of some population of problems (e.g., calculus problems), it is useful to estimate an LLM's TTV performance as a function of $M$, which can be used to find a suitable tradeoff between performance and computational cost, or optimize hyperparameters (e.g., Temperature) in pursuit of a more favorable overall tradeoff.

In the literature, however, evaluating the TTV of an LLM is currently done by applying the LLM to each prompt $M$ times (for varying settings $M$), and then applying TTV to the responses in each case. Typically, in the literature $M \in [20, 256]$ (Lewkowycz et al. (2022); Wang et al. (2023);Gemini Team & et al. (2024);Anthropic (2023);Quach et al. (2024);Chen et al. (2024);DeepSeek-AI & et al. (2025);Lightman et al. (2023);Uesato et al. (2022)), therefore requiring at least that many LLM evaluations for *each test prompt*, and many test prompts may be needed to obtain a representative

estimate of TTV for a population of prompts (e.g., math problems). This process may also need to be repeated multiple times for different hyperparameter settings. In practice this approach is problematic because it requires such a large number of LLM evaluations, and evaluating an LLM just once is already computationally intensive. This makes it costly or even prohibitive to evaluate the performance-computation tradeoff of TTV for LLMs, or to optimize their hyperparameters.

**Contributions of this Work.** In this work we propose two methods to estimate the voting-accuracy of an LLM with substantially less computational cost than current methods. Our approach relies upon using a relatively small number, $G$, of LLM evaluations for each test prompt - we show that $G = 5$ is often sufficient - which are used to characterize the probability of sampling correct and incorrect responses from the LLM. Using this information, we show that one can efficiently estimate the accuracy of an $M$-sized test-time ensemble, denoted $p^*(M)$. We propose two approaches: one based upon a Monte-Carlo estimator, and another approach based upon a Gaussian approximation assumption. Using a popular public benchmark that is frequently assessed using voting accuracy (e.g., MATH Hendrycks et al. (2021)), we demonstrate that our two estimation approaches can closely approximate the true ensemble accuracy, while requiring substantially less computation than a purely empirical approach, especially as $M$ grows larger. We also investigate key assumptions upon which our estimators rely, and how our method respond to changes in hyperparameters like temperature. We release a set of Python-based functions that can be used to implement our methods in our supplementary material.

## 2 RELATED WORK

Most existing research on voting, or TTV, is focused upon developing novel voting methods to improve the final accuracy of LLMs when applied to a specific task (Lewkowycz et al. (2022); Wang et al. (2023);Gemini Team & et al. (2024);Anthropic (2023)). Some other works develop new methods to scale model performance using new voting methods (Chen et al. (2024)), early exiting procedures (Chen et al. (2024);Quach et al. (2024)), or reinforcement learning ensemble approaches (Lightman et al. (2023);Uesato et al. (2022)). By contrast, this work proposes a computationally efficient method to *estimate the accuracy of particular TTV scheme - specifically the widely-used plurality and majority vote methods*. The motivation for developing efficient accuracy estimation methods for TTV has only emerged very recently, due to the success of TTV with LLMs, and the substantial computational cost of evaluating LLMs. To our knowledge, the only existing approach used to estimate the accuracy of TTV is the purely empirical discussed in Sec. 1, which is widely used in the literature (Lewkowycz et al. (2022); Wang et al. (2023);Gemini Team & et al. (2024);Anthropic (2023);Quach et al. (2024);Chen et al. (2024);DeepSeek-AI & et al. (2025);Lightman et al. (2023);Uesato et al. (2022)).

## 3 MOTIVATION & PROBLEM SETTING

Here we describe the motivation and problem of estimating the accuracy of LLMs, and especially LLM ensembles. ***To aid the reader's understanding our methodology we provide a notation table in Appendix A***.

### 3.1 MOTIVATION

Accuracy and voting accuracy are related but distinct: two models can have the same accuracy yet very different voting accuracy. Figure 1 illustrates this with a synthetic test set. Panel (a) shows the per-question success rates $(p_n^*)$ for two LLMs. Although the distributions differ, their means—i.e., standard accuracy when each question is asked once—are identical. Panel (b) shows the consequence for voting: as we increase samples per question from 1 to 100 and take a majority vote, LLM 1 converges to a lower voting accuracy than LLM 2. Standard accuracy alone therefore fails to predict behavior under voting; estimating voting performance requires multiple responses per prompt.

This distinction has compute implications. Each LLM inference is costly: even small models (e.g., Llama 3.2 1B) require $\sim 2.46$ TFLOPs per query, while very large models (e.g., Llama 3 405B) can exceed $\sim 810$ TFLOPs per query (Grattafiori (2024)). Repeated sampling for voting can strain

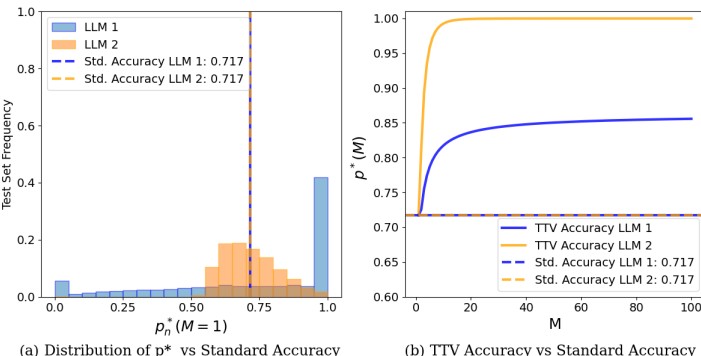

(a) Distribution of $p_n^*$ vs Standard Accuracy     (b) TTV Accuracy vs Standard Accuracy

Figure 1: Illustration of Accuracy vs Majority Vote Accuracy using two hypothetical LLMs with the same accuracy, but very different ensemble accuracies. (a) A histogram of the probability of returning the correct response for individual questions. (b) A comparison of the same models using voting-accuracy at different values of $M$, where $M$ represents the number of responses in the vote ensemble. $p_n^*(M)$ is the accuracy of the LLM on prompt $n$ given $M$ repetitions of prompt $n$, and $p^*(M)$ is the overall voting-accuracy of the LLM averaged across all prompts in the test set.

even well-resourced organizations, so methods that estimate voting accuracy with fewer queries are prudent.

### 3.2 PROBLEM SETTING

Here we present a rigorous description of plurality voting with LLMs, as well as the current empirical method of evaluating their accuracy. We assume that we have some model (e.g., an LLM, $f$) with parameters $\phi$ that takes a user-provided prompt $X \in \mathcal{X}$ as input and returns a random sequence of tokens, $Y$, and given by

$$Y = f_{\phi,t}(X, Z) \sim P_{Y|X} \tag{1}$$

Where, $X \sim P_X$ is a distribution over prompts, $Z \sim P_Z$ represents the randomness of the LLM even when given a fixed input prompt, which arises due to sampling output tokens from the LLM (e.g., we sample each output token based upon probabilities produced by the LLM). The distribution over $Y \in \mathcal{Y}$ is denoted $P_{Y|X}$. Here the subscript $t$ is some user-provided parameter that influences the properties of $P_{Y|X}$, such as the "Temperature" parameter that is common in LLMs.

$M$-**Plurality Ensembles.** This is a widespread approach to improve the accuracy of LLMs, where we present some specific prompt $X = x$ to the LLM repeatedly $M$ times and then identify the output of the LLM that occurs most frequently[1]. Mathematically, we set $X = x$ and draw $M$ samples $\{Y_m(x)\}_{m=1}^M \sim P_{Y|X=x}$, where $P_{Y|X=x}$ denotes the distribution that results from setting $X = x$. Note that $M$ is not a random variable. Then we count the number of instances of each candidate solution

$$C_M(y, x) = \sum_{m=1}^M \mathbb{1}[Y_m(x) = y], \forall y \in \mathcal{Y} \tag{2}$$

The quantity $C_M(y, x)$ is a random variable that indicates the number of times that we encounter output $y$, given $M$ repeated prompts of our LLM with $x$. Then the output of the ensemble is the output that appears most frequently, given by

$$Y_M(x) = \arg\max_y C_M(y, x) \tag{3}$$

---

[1]When scoring LLMs, it is common to create a "grading" function that identifies analogous LLM output and treats them as a single candidate solution.

Here $Y_M(x)$ is a random variable representing the *single* output returned by the $M$-member plurality vote from the LLM, given a particular prompt $x$. Although $x$, $y$, and $M$ are not random quantities there is randomness due to $Z$ from Eq. 1 that generates each individual output in the ensemble.

**The Problem: Accuracy Estimation.** In this work we consider problems where there is some member $Y^*(X) \in \mathcal{Y}$ that represents the desired output of the LLM, which depends upon the prompt, $X$. A widely-used performance metric is accuracy, given by

$$p^*(M) = (1/M)\mathbb{E}_{\sim X,Z}[\mathbb{1}[Y_M(X) = Y^*(X)]] \tag{4}$$

which is typically measured over some distribution of prompts $P_X$ and some scheme for sampling different LLM outputs when $X = x$, $P_Z$. The accuracy $p^*(M)$ can also be interpreted as the average probability of producing the correct response over some population of prompts. To approximate $p^*(M)$ we can sample prompts $\{x_n\}_{n=1}^N \sim P_X$ and then sample plurality votes for each prompt $\{y_{k,M}(x_n)\}_{k=1}^K \sim P_{Y_M|X=x_n}$. In other words, for each prompt, $x_n$, we create $K$ independent $M$-sized ensembles, where $y_{k,M}$ represents the candidate chosen by the $k^{th}$ ensemble. We then compute the sample accuracy estimator

$$\hat{p}^*(M) = \frac{1}{NK} \sum_{n=1}^N \sum_{k=1}^K \mathbb{1}[y_{k,M}(x_n) = y^*(x_n)] \tag{5}$$

Although this estimator is widely used in the literature, it is computationally expensive, requiring $NKM$ calls to an LLM. For simplicity of exposition, our subsequent discussion focuses on estimating plurality vote accuracy for a single prompt, $x$, so that $N = 1$ and we have

$$\hat{p}^*_{emp}(M) = \frac{1}{K} \sum_{k=1}^K \mathbb{1}[y_{k,M}(x) = y^*(x)] \tag{6}$$

We can always recover the sample estimator in Eq. 5 by averaging $\hat{p}_{emp}(M)$ over a representative sample of prompts.

## 4 Proposed Accuracy Estimators

We propose two methods to estimate plurality ensemble accuracy, which each trade-off computationally efficiency and estimation accuracy (of the LLM's TTV accuracy) to varying degrees. We argue that both methods have preferable tradeoffs compared to the empirical estimation (Eq. 6).

### 4.1 Monte-Carlo Estimator, $\hat{p}^*_{mc}$

Our approach relies on the assumption that, given a fixed input prompt $X = x$, the output of LLMs, $Y(x)$, are independent and identically distributed (iid). This condition holds if $P_Z$ is iid and $f_\phi$ is deterministic, and both of these conditions are generally satisfied in practice. Given these assumptions, the output of an LLM can then be treated as a categorical distribution over possible outcomes, $Y(x) \sim P_{Y|X=x} = CAT(\mathbf{p}(x))$, where $\mathbf{p}(x) = \{p_y(x)\}_{y=1}^{|\mathcal{Y}|}$, where each $p_y(x)$ is the probability of sampling $y \in \mathcal{Y}$, conditioned on prompt $x$. It then follows that the number of votes for each candidate solution returned after $M$ prompts (e.g., from a plurality vote) is multinomial, $C_M(y, x) \sim \text{MULTI}(\mathbf{p}(x), M)$.

Therefore, if we know $\mathbf{p}(x)$ then we can construct a computationally inexpensive Monte-Carlo estimator for $p^*(M)$ by sampling votes $\mathbf{c}_M(x) = \{c_M(y, x)\}_{y \in \mathcal{Y}} \sim \text{MULTI}(\mathbf{p}(x), M)$. From here we can proceed to estimate one sample estimate of $p^*(M)$ using Eq. 3 to obtain a single plurality vote sample. We can then repeat this process $K$ times and use the collection of samples in Eq. 6. Since we are sampling from a multinomial distribution rather than from an actual LLM, these operations are substantially more computationally efficient, however, we need some way to estimate $\mathbf{p}(x)$.

Our proposed solution involves drawing a relatively small number of $G$ sample outputs from the LLM, $f_{\phi,t}$, with which we can estimate $\mathbf{p}(x)$ by counting the votes for each output candidate, $c(y, x)$

as shown in Eq. 2, but with $G$ samples instead of $M$. Crucially though, instead of proceeding to estimate a plurality vote outcome as before, we use these data instead to obtain a sample estimator $p_y(x) \approx \hat{p}_y(x) = \frac{1}{G}c(y, x)$. We can then use the Monte-Carlo approach outlined above to estimate $p_M^*$, and we denote the resulting estimator as $\hat{p}_{mc}^*$.

*Advantages and potential limitations.* Although the monte-carlo approach still requires obtaining samples from an LLM, we anticipate that generally $G << KM$ evaluations required by the widely-used empirical estimator in Eq. 6. For example, whereas $M$ grows in proportion to the size of the plurality ensemble, $G$ will generally remain fixed, giving it better scaling properties. Furthermore, sampling from a Multinomial is orders of magnitude faster than evaluating an LLM, and therefore we can run a very large number of plurality vote simulations (i.e., $K$ in Eq. 6) in real time at very little computational cost.

Because Monte-Carlo will allow for a very large $K$, it is likely to converge to a highly accurate solution of $p_M^*$, assuming our estimate of $\mathbf{p}(x)$ is accurate. However, because our estimate of $\mathbf{p}(x)$ will always be imperfect, Monte-Carlo will also converge to an imperfect solution. One goal of our numerical experiments will be to estimate this error. Also, in principle the space of LLM outputs will be much larger than the number of LLM samples we collect: i.e., $|\mathcal{Y}| \gg |G|$. Therefore, with just $G$ samples, we will not encounter most possible LLM outputs in $\mathcal{Y}$, but in practical scenarios most of these outcomes will have negligible probabilities, and crucially, only a few outputs (often including the correct solution) will occupy most of the probability mass, so that $G$ can be quite small. As we show in our experiments, for most scenarios, $G = 5$ is suitable for estimating $p^*(M)$ for any size of $M$.

## 4.2 Analytical Estimator, $\hat{p}_{ana}^*$

Here we develop an analytical estimator for $p^*(M)$ as well, so that we can estimate it for any value of $M$ using a closed-form expression. We begin by observing that $p^*(M)$ can be expressed as the probability that there are more votes for the true solution than for every other unique candidate solution returned by the LLM. Mathematically we can express this as

$$p^*(M) = Pr[\cap_{y \in \mathcal{Y} \setminus y^*} [C_M(y^*, x) > C_M(y, x)]] \tag{7}$$

However, each $C_M(y, x)$ is a random variable sampled from a shared Multinomial distribution and, to our knowledge, there is no known analytical expression for this quantity. We propose to overcome this limitation by approximating each $C_M(y, x)$ in Eq. 7 as independent and normally distributed with their own mean and variance. Indeed, based on the central limit theorem (Yates & Goodman (2014)), when $C_M(y, x)$ is sufficiently large, it can be approximated as a normal distribution, which we denote $V_M(y, x)$, with mean, $\mu(y)$, and variance, $\sigma^2(y)$ given by

$$\mu(y) = Mp_y(x), \text{ and } \sigma^2(y) = Mp_y(x)(1 - p_y(x)) \tag{8}$$

where as earlier, $p_y(x)$ is the probability of sampling outcome $y$ from the LLM. Therefore, like the Monte-Carlo estimator, this approach requires an estimate of these probabilities, and again we use $G$ samples to estimate each $p_y(x)$ for the Normal approximation.

Using the Normal approximation we can treat each random variable as independent and we can replace the conjunction in Eq. 7 with a product, given by

$$\hat{p}_{ana}^*(M) = \prod_{y \in \mathcal{Y} \setminus y^*} Pr[V_M(y^*, x) > V_M(y, x)] \tag{9}$$

where we have replaced $C_M(y, x)$ with its normal approximation $V_M(y, x)$. Each term in the product in Eq. 9 is now an inequality between two independent normal random variables, which has a well-known analytic expression, given by

$$Pr[V_M(y^*, x) > V_M(y, x)] = \Phi\left(\frac{\mu(y^*) - \mu(y)}{\sqrt{\sigma^2(y^*) + \sigma^2(y)}}\right) \tag{10}$$

*Advantages and potential limitations.* In similar fashion to the Monte-Carlo estimator, the analytical approach here uses a sample estimator for the probabilities of each LLM output, $p_y(x)$, given a

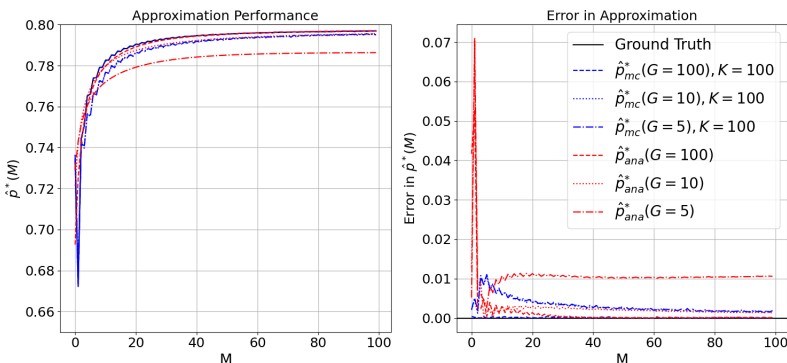

Figure 2: Comparison of Approximation Method Performance for GPT-4o Mini on the MATH dataset. $M$ represents the number of responses in the TTV ensemble, and $p^*(M)$ is the overall TTV accuracy of the LLM averaged across all prompts in the test set.

specific prompt. However, whereas the Monte-Carlo approach relies on these probabilities to sample plurality vote outcomes, the approach here uses them in a closed-form approximation to $p^*(M)$, which requires substantially less computation. However, this approach relies upon a Normal approximation which may introduce varying degrees of error in the final estimate of $p^*(M)$. The Normal approximation works best when there are relatively few non-zero $C_M(y, x)$ values, and they are relatively large, so that the central limit theorem leads to a good Normal Approximation.

## 5 EXPERIMENTS

We constructed experiments to analyze three key aspects of our methodology:

1. Close approximation to empirical results, section 5.1.

2. Our approximations result in an substantial reduction in required compute and processing time (e.g., they are efficient), section 5.2.

3. Verify key assumptions, section 5.3. Here we verify key assumptions (namely that only a few outputs will occupy most of the probability mass, so that $G$ can be quite small)

We evaluated our methods on a standard LLM benchmark that is routinely used in voting-accuracy studies: MATH (Hendrycks et al. (2021)). We tested two models: a commercial model, GPT-4o-mini (gpt-4o-mini-2024-07-18) (OpenAI (2024)), and an open-source models, LLaMA-3.2 1B (Instruct BF16) (Grattafiori (2024)). We accessed GPT-4o-mini via OpenAI's Python API.

Our experiments used prompting practices common in related litterature—specifically, zero-shot chain-of-thought (Kojima et al. (2022); Wei et al. (2022)). We did not study few-shot or other prompts because (1) they are not necessary to assess our methodology, (2) API costs for large-scale runs on state-of-the-art models are non-trivial, and (3) the compute required for extensive prompt sweeps is likewise non-trivial.

For our open source model we used a 10% sample of the test set, since full-set evaluation is unnecessary. This is because our goal is to inform developers about expected estimation ability on a set of problems—not to establish a competitive baseline—so a representative sample suffices. Moreover, each prompt was queried 50 times per model, making full-set runs prohibitively resource intensive.

We examine the performance characteristics of our two accuracy estimators ($\hat{p}_{ana}$ and $\hat{p}_{mc}$) for LLM ensembles of sizes $M \in [10, 100]$. We choose this range because it is representative of what is generally used in practice, and sufficiently large to demonstrate the efficacy of our methods. To evaluate the accuracy of our two accuracy estimators, we need the "true" accuracy of our LLMs for our testing dataset for each setting of $M$. In practice however we can only approximate the true accuracy of an LLM ensemble. Obtaining a high quality accuracy estimate using existing empirical methods is highly computationally costly: potentially requiring predictions from several instantiated $M$-sized ensembles for each question. In principle, our proposed monte-carlo estimator, $\hat{p}_{mc}$ is

| Method | Time (Compute hours) | TFLOPS |
|--------|----------------------|--------|
| Empirical | 1351.1 | 1,230,000.0 |
| Monte-Carlo | 1.25 | 0.0003 |
| Gaussian | 0.017 | 0.000003 |

Table 1: Time & Compute Comparison of Methods for Llama 3.2 1B Instruct BF16 on the MATH Benchmark. Empirical results use $G = 100$; Monte-Carlo Plurality-vote results use $G = 10$, $M = 100$, $K = 100$; Gaussian Plurality-Vote use $G = 10$, $M = 100$.

highly accurate as long as $G$ and $K$ are sufficiently large. Therefore, we obtain ground truth accuracy for each problem in our dataset by drawing $G = 50$ to 100 (depending on the specific experiment) samples from the LLM, and then running $K = 1000$ monte-carlo simulations to estimate accuracy. We do this for $N = 500$ or 5000 prompts in our test dataset, resulting in ground truth accuracy estimates for $N = 500$ or 5000 problems, and for all values of $M \in [10, 100]$.

## 5.1 Approximation of Empirical Results have low Error

To evaluate the precision of our methods we evaluated $N = 5000$ prompts 100 times each (e.g., $G = 100$) from the MATH dataset using a commercial class model, ChatGPT 4o-mini. We accessed this model via OpenAI's python based API, and used it's default hyperparameter settings (namely temperature = 1.0).

Our experiments indicate that our plurality-vote estimation methods approximate empirical plurality-votes with very low error using very limited sample sizes. This is shown in figure 2, where both our Monte-Carlo and analytical estimator are able to approximate the truth with $\sim 1\%$ error when only using $G = 5$. In Figure 2 $M$ was varied from 1 to 100, showing that our methods completely characterize this plurality-vote performance with minimal compute. Furthermore, in Figure 2 we see that increasing $M$ beyond 5 samples has little value (in Figure 2, and that the lines for a plurality vote using $M = 10$ and $M = 100$ nearly overlap). *This indicates that our method can precisely and efficiently approximate the maximum performance that can be obtained* from a language model through response ensembling.

## 5.2 Approximation of Empirical Results are Efficient

Our results, shown in table 1, indicate that using our methods can save a substantial amount of time (several orders of magnitude) and compute when evaluating a model's TTV accuracy on a dataset.

To evaluate the efficiency of our method we evaluated $N = 500$ prompts 50 times each (e.g., $G = 50$) from the MATH data set using an open source model (Llama 3.2 1B Instruct BF16 using default hyperparameter settings, and temperature=1.0). We ran this model on a single NVIDIA RTX-TITAN with 24GB of VRAM using the Huggingface Python transformer pipeline.

To conduct this efficiency experiment we calculated the average amount of time needed to generate a response to 1 prompt. This average turned out to be 9.728 seconds, and we used this average to determine how long it would take to complete a full plurality vote experiment of the MATH dataset like those commonly found in the literature. Specifically, in table 1 we show how long it would take to evaluate all $N = 5000$ MATH dataset questions $G = 100$ times each empirically. In table 1 we compare this to our approximation methods with $G = 10$, $M = 100$, and $K = 100$ for our Monte-Carlo method. Our approximation methods were run on a MSI GL65 Leopard with a Intel Core i7 10th Generation CPU in Python.

We also provide a rough assessment of the amount of compute that will be used by each method under the same conditions. Our empirical compute estimate was generated according to Kaplan et al. (2020) and Grattafiori (2024) which detail the number of FLOPS used to provide a response to a prompt. Notably, in our estimates LLama 3.2 1B Instruct BF16 uses 2.46 TFLOPS to respond to a prompt. We used OpenAI's GPT-o3 model to estimate the number of FLOPS our Monte-Carlo and analytical estimator code used.

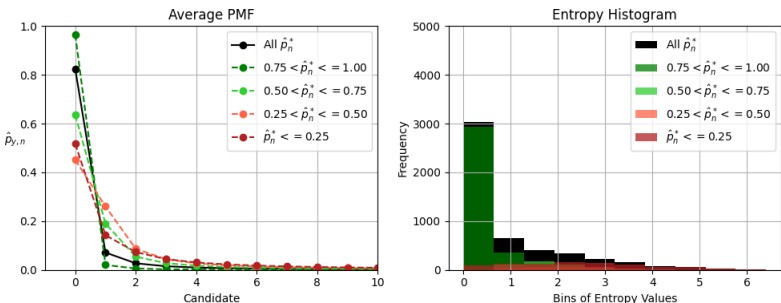

Figure 3: Probability Mass Function and Entropy Analysis for GPT-4o-Mini on the MATH dataset. $\hat{p}_{y,n}$ is the estimated probability of observing candidate $y$ for prompt $n$. $\hat{p}_n^*$ is the estimated probability of observing the correct candidate on prompt $n$.

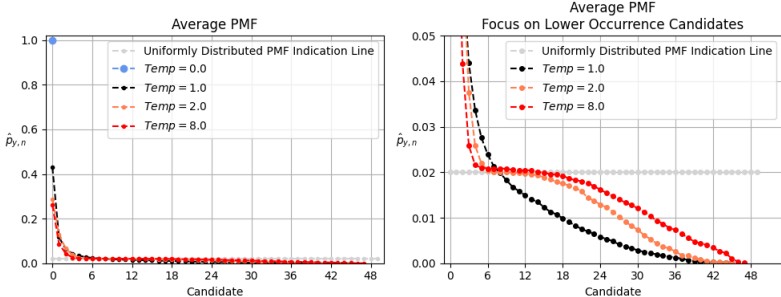

Figure 4: The Effect of Temperature on Answer Candidates PMF Distribution for Llama 3.2 1B Instruct on the MATH Dataset

## 5.3 FURTHER ANALYSIS

**Verification of Key Assumption**. Our plurality-vote methodologies assume the majority of the probability mass for any particular prompt are concentrated in a small set of answer candidates. In figure 3 we present analysis of GPT-4o-mini ($Temperature = 1.0$) on the MATH dataset where we draw $M = 100$ samples of $N = 5000$ prompts, and we show that on average *this assumption holds*.

In our analysis we assumed that: the full set of candidates for each prompt has 100 members (e.g., $\mathbf{p}(x)$ has 100 entries), and that if we did not actually observe 100 different answer candidates for any particular prompt that many of the candidates had such a small rate of occurrence that it was acceptable to say that $C_{M=100}(y, x) = 0$. Given these assumptions, figure 3 shows that for GPT-4o-mini responses to the MATH dataset, on average the probability mass is almost completely concentrated in 10 candidates or less, and that on average 95% of the probability is concentrated in 5 candidates. Figure 3 also shows that this holds even for the questions that the model finds difficult (e.g., where $\hat{p}_n^* \leq 0.25$ ). Additionally, figure 3 shows that the entropy of each prompt is low, further indicating that our central limit theorem assumption holds.

**Effects of Temperature on our Methods**. To explore the effects of temperature on our methodology we used the Llama 3.2 1B Instruct model and the MATH dataset (we used an open source model for these experiments to help control cost). Due to the generation time per prompt iteration (average 9.78 seconds, when running the model on an NVIDIA RTX Titan) we only used 500 of the 5000 total prompts (e.g. $N = 500$), and we conducted $M = 50$ iterations of each prompt. Our sample of the original 5000 questions was stratified across the different topics and problem difficulty levels specified in the original data set.

Our temperature experiments indicate what we expect to see with respect to the candidates generated by a LLM as temperature changes: as temperature increases the number of candidates that the model generates also increases (at temperature=0.0, there is only one candidate, and as temperature increases more candidates arise), and the distribution of the probability mass becomes more uniform (shown in figure 4). Subplot b of figure 4 shows that as temperature increases, each candidate's share

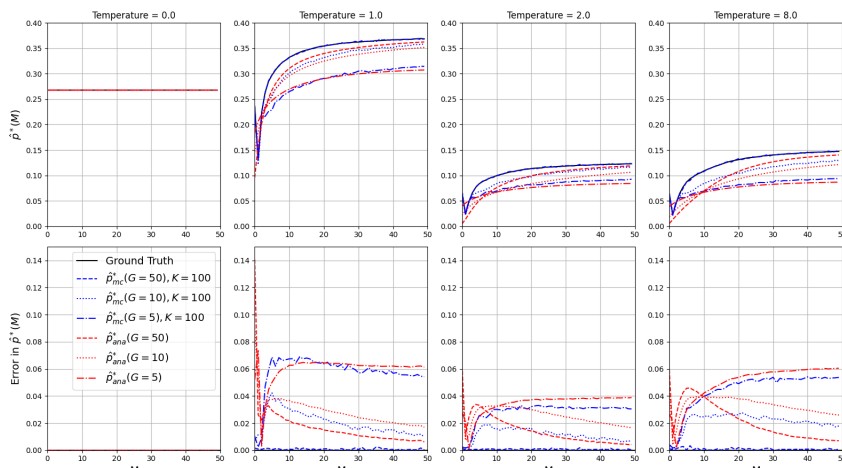

Figure 5: The Effect of Temperature on Approximation Error for Llama 3.2 1B Instruct on the MATH Dataset. $\hat{p}_{y,n}$ is the estimated probability of observing candidate $y$ for prompt $n$.

of the probability mass begins to approach the amount we would expect in a uniform distribution (e.g., 0.02).

In figure 5 we show that our methods' ability to approximate the ground truth still holds as temperature increases. We see that, as we would expect, our methods perfectly approximate the ground truth when temperature=0.0 (because there is only one answer variety), and that for all temperature settings our maximum approximation error is $\sim 7\%$ when $M \geq 5$. Notably we show this approximation error can be reduced to $\leq 5\%$ by setting $G = 10$. We hypothesize that the approximation error is lower for model's with $temperature \geq 2.0$ in these experiments because the model has such a low probability of answering of observing the correct candidate (on average $\leq 0.15$ for both the $temperature = 2.0$ and $temperature = 8.0$ models).

## 6 LIMITATIONS

Here we only explore how model behavior changes, and how well our methods approximate model behavior, as a function of temperature, and not top_p or other LLM hyperparameters. Our experiments include extreme sizes of LLMs (small, and large) but no mid-sized models due to compute limitations. Our experiments include only one common benchmark dataset, again due to compute and budget limitations, but we believe that in spite of this the general principle we have demonstrated (model performance at larger TTV ensemble sizes can be precisely estimated using a smaller ensemble) holds. We do not develop methods to estimate the performance of newer ensembling methods like filter-vote (Chen et al. (2024)) or other reinforcement learning practices (Lightman et al. (2023); Uesato et al. (2022)). If these new methods gain similar traction to majority has, developing compute efficient evaluation methods will be a prudent avenue of research.

## 7 CONCLUSION

Language models are increasingly assessed using a voting approach, but this methodology is extremely computationally intensive. We presented several performance estimation methods that character maximum possible LLM performance using voting approaches, and showed that the assumptions necessary for these methods hold under the standard use case conditions. Furthermore, we showed that our methods are efficient and precise because they have very low error, and require very limited time and compute. Our research shows that our methods can precisely approximate the results of a large empirical evaluation (100 trials per prompt) using a comparatively small sample size ($5 - 10$ samples per prompt), resulting in a reduction of test time hours and compute TFLOPS by 8 to 11 orders of magnitude.

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

# A  NOTATION

Table 2: Notation Table

| Notation | Description |
|---|---|
| $f$ | Large language model (LLM) like GPT-4o-mini, LLaMa-3.2 3B Instruct, Gemma-3-1b-it, etc. |
| $t$ | LLM hyperparameter that influences the entropy (e.g., Temperature) |
| $\phi$ | Parameters of a LLM |
| $N, n$ | The number of problems in a language model test set; indexed by $n$. Note, this is **not** a random variable, it is specified. |
| $M, m$ | The number of queries in an input ensemble, $M$; indexed by $m$. Note, this is **not** a random variable, it is specified. |
| $K, k$ | The number of sample sets draw to produce an estimate; indexed by $k$. Note, this is **not** a random variable, it is specified. |
| $G, g$ | A sample draw of queries used to estimate model performance with $M$, or $KM$ queries; indexed by $g$. We expect that $G < M$, and $G << KM$. Note, this is **not** a random variable, it is specified. |
| $*$ | The truth/correct-output indicator |
| $\mathcal{X}$ | The input space of all possible valid prompts that can be given to the model. |
| $X$ | A random variable representing a random draw from the input space |
| $x, x_n$ | A specific prompt. $x_n$ specifies the index of the specific prompt in the dataset |
| $\{x_n\}_{n=1}^N$ | The dataset of specific prompts used to evaluate the model |
| $P_X$ | The probability distribution of the random variable $X$. This specifies how likely each $x \in \mathcal{X}$ is. |
| $\tilde{\mathcal{Y}}$ | The space of all latent/raw outputs — possibly logits, token sequences, internal states, etc. |
| $\mathcal{Y}$ | Set of possible final outputs (e.g., strings, labels) |
| $\tilde{Y}$ | Random variable of latent output from LLM |
| $Y$ | The final, observable output random variable — what the user or evaluator actually sees. |
| $y, y_n$ | A specific output from the model, $y_n$ specifies the index of the specific prompt in the input dataset that resulted in $y_n$ |
| $Y_M$ | A random variable representing $M$ draws from a LLM $f$ for each input |
| $Y_M(x)$ | random variable representing the single output returned by the M-member plurality vote from the LLM, given prompt $x$ |

| Notation | Description |
|---|---|
| ...continued from previous page | |
| $\{Y_m(x)\}_{m=1}^M$ | A set of $M$ LLM responses to input prompt $x$ |
| $Y_m$ | The $m$-th draw of random variable $Y_M$ |
| $P_{Y|X=x}$ | The distribution of $\{Y_m(x)\}_{m=1}^M$ given $X = x$ |
| $C_M(y, x)$ | The count of occurrences for candidate $y$ in response to input $x$, given $M$ repeated prompts (e.g., an $M$ member plurality) |
| $c(y, x)$ | The count of occurrences for candidate $y$ in response to input $x$ |
| $\mathbf{c}_M(x)$ | The count of occurrences for each candidate in response to input $x$, given an $M$ repeated prompts |
| $MULTI(\mathbf{p}(x),\ M)$ | The **multi**nomial distribution of votes for each candidate stemming from input $x$ (e.g., $C_M(y, x) \sim MULTI(\mathbf{p}(x),\ M)$) |
| $CAT(\mathbf{p}(x))$ | The **cat**egorical distribution of possible outcomes stemming from input $x$ (e.g., $Y(x) \sim CAT(\mathbf{p}(x))$) |
| $Y^*(X)$ | The desired, or correct, output of the LLM given input $X$ |
| $y^*$ | The specific correct output of the LLM |
| $P_{\tilde{Y},X}$ | The distribution of latent outputs, $\tilde{Y}$, given input $X$ |
| $P_{Y,X}$ | The distribution of final outputs, $Y$, given input $X$ |
| $h$ | Surjective "onto" mapping function from $\tilde{Y} \rightarrow Y$, e.g. $Y = h(\tilde{Y})$ |
| $Z$ | Represents the randomness of the model even when given a fixed input prompt, which arises due to sampling output tokens from the model |
| $P_Z$ | The distribution of the internal randomness of the model |
| $p_n^*(M),\ \hat{p}_n^*(M)$ | The probability of observing the truth candidate (and it's estimate ) for prompt $n$ given $M$ trials. The *estimated* probability of of observing the truth candidate prompt $x_n$ given $M$ trials. |
| $p^*(M),\ \hat{p}^*(M)$ | The accuracy (and it's estimate) of the LLM given $M$ repeated queries of each prompt (e.g., the Test-Time-Voting (TTV) accuracy of the model, or the average probability of producing the correct response over some population of prompts). |
| $\hat{p}_{emp}^*(M)$ | The empirical estimate of $p^*(M)$ |
| $\hat{p}_{mc}^*(M)$ | The Monte-Carlo estimate of $p^*(M)$ |
| $\hat{p}_{ana}^*(M)$ | The analytical (e.g., Gaussian) estimate of $p^*(M)$ |
| $\mathbf{p}(x)$ | A vector containing the probability of observing each candidate in response to input $x$, e.g., $\{p_y(x)\}_{y=1}^{|\mathcal{Y}|}$ |
| $p_y(x)$ | The probability of observing candidate $y$ for prompt $x$ |

| ...continued from previous page | |
|---|---|
| **Notation** | **Description** |
| $V_M(y, x)$ | The number of 'votes' for candidate $y$ given input prompt $x$ is repeated over $M$ queries. |
| $\mu(y)$ | Mean votes for candidate $y$ for prompt $x$ (e.g., mean of r.v. $V_M(y, x)$) |
| $\sigma^2(y)$ | Variance in votes for candidate $y$ for prompt $x$ (e.g., variance of r.v. $V_M(y, x)$). |

## B    CODE

The python code for our approximation functions is included as .py in our supplementary material.

## C    SYNTHETIC EXAMPLE DETAILS

The synthetic example we present in Section 2.1 was specifically designed to show that two datasets can have the same accuracy but different ensemble accuracy. To construct this example we randomly generated one data distribution of 10000 $p_n^*$ values between 0 and 1 with a fixed standard deviation of 0.1, and mean of 0.717. We then generated a second data distribution of another 10000 $p_n^*$ values between 0 and 1 with a fixed standard deviation of 0.35, and mean of 0.717. For each of these two distributions, we then generated $p_y$ for four additional candidates

## D    LLM USAGE

We used a LLM, GPT-4o, to assist in retrieval and discovery of related works.

