# OpenReview forum: "Compute-efficient Evaluation of LLM Voting Accuracy"
_ICLR.cc/2026/Conference — ICLR 2026 Conference Withdrawn Submission_

### Official Review · Reviewer_Ccxh · 2025-10-25

**Soundness:** 2
**Presentation:** 1
**Contribution:** 1
**Rating:** 2
**Confidence:** 4

**Summary:**

This paper proposes two computationally efficient methods to accurately estimate the voting accuracy of Large Language Models (LLMs).

**Strengths:**

The research topic is important.

**Weaknesses:**

- The proposed methods suffer from significant technical flaws that undermine the paper's contributions.

  1. The first proposed method is simply an application of the parametric **bootstrap**. The method first estimates the parameters of the underlying categorical distribution, and then uses this estimated distribution to run Monte-Carlo simulations. This is a well-known, textbook procedure. The methodological novelty here is therefore minimal.

  2. The second proposed method is based on a **fundamentally flawed assumption**. Specifically, Eq. (10) is based on the assumption that $V_M(y^*,x)$ and $V_M(y,x)$ are all independent. This assumption is false. They are, in fact, negatively correlated: a higher count for correct solutions necessarily implies a lower total count for all other candidates. The correct way is to apply the *multivariate* central limit theorem to the vector $C_M(\cdot, x)$ and approximate it with a multivariate normal distribution. This distribution would have a non-diagonal covariance matrix that captures the negative dependencies between the counts. A correct (consistent) estimation should take the covariance into account.

- The paper also has many presentation issues, detracting from the paper's professionalism.
  1. Line 15: I don't think $M$ should be defined in the abstract.
  2. Line 32: Incorrect quotation marks.
  3. The paper uses "majority voting", "plurality voting", and "plurality ensembles" interchangeably. I recommend the authors to make this consistent and stick to the most commonly-used "majority voting".
  4. Eq, (4): Why is $1/M$ needed here?
  5. The paper is inconsistent in its presentation of the "temperature" hyperparameter. It appears variously as "Temperature" (capitalized), "temperature" (lowercase), and at times is _italicized_.
  6. Line 221: $<< \to \ll$ (`\ll` in latex).

**Questions:**

- In Table 1, do the TFLOPS results of the proposed method include the cost for sampling $G$ solutions?

---

### Official Review · Reviewer_fSjZ · 2025-10-26

**Soundness:** 1
**Presentation:** 2
**Contribution:** 1
**Rating:** 2
**Confidence:** 4

**Summary:**

The paper proposes a method to estimate the accuracy of **test-time voting** for LLMs using only a small number of samples. Instead of running many generations to measure majority-vote accuracy empirically, the method samples $G$ outputs per question, estimates the output distribution, and uses Monte Carlo simulation to predict accuracy under a larger vote size $M$.

Experiments on the **MATH dataset** with **GPT-4o-mini** and **Llama-3.2 1B Instruct** compare predicted and empirical accuracies across different sampling temperatures and vote sizes. The results suggest that the proposed estimator can approximate full-sampling accuracy using significantly fewer model queries.

**Strengths:**

- The motivation, reducing the cost of majority-vote inference while maintaining accuracy estimates, is easy to understand and potentially useful if the method worked reliably.
- Despite a limited scope, the experiments (especially in Figure 2) illustrate how accuracy saturates with the number of samples.

**Weaknesses:**

- The paper assumes that the empirical distribution of LLM outputs can be reliably estimated with a very small number of samples (e.g., $G = 5$ or $10$).
This assumption is not justified either theoretically or empirically. In practice, LLM output distributions are often highly multimodal and long-tailed.
- Although the paper claims low approximation error, its “ground truth” accuracy is itself obtained from Monte Carlo simulations rather than actual LLM ensemble runs, making the validation self-consistent but not empirically independent. (Line 316, 335-336).
- Table 1 underestimates the computation for the Monte-Carlo and Gaussian methods by excluding the FLOPs of the required $G$ LLM queries (see Sec. 4.1), while the empirical baseline includes LLM inference FLOPs. Using the paper’s own 12,300 TFLOPs/query, $G$ queries would add ~$12,300G$ TFLOPs to Monte-Carlo, which is orders of magnitude larger than the values reported.
- The errors in Figure 8 are already substantial, and the inclusion of $T = 2$ and $T = 8$, temperatures that are effectively nonsense in practical LLM inference, further undermines the credibility of the analysis. The authors should focus on realistic temperature ranges (e.g., $T \in [0, 1.5]$) and discuss why their estimator’s performance deteriorates outside that regime.
- The curves are not clear enough in Figures 2 and 5.
- The experimental validation is extremely narrow. The authors only test their method on **two models** (GPT-4o-mini and Llama-3.2 1B-Instruct) and a **single dataset (MATH)**. This limited scope raises concerns about the generality of the proposed estimator.
- Terminology:
    - (Line 32-33) Actually, TTV is not a well-known acronym in LLM inference literature, and I prefer to use the full name (test-time voting), answer aggregation, etc.
    - (Line 33-34) In the most recent LLM literature (e.g., [1][2][3]), *“majority voting”* is used colloquially to mean selecting the **most frequent** answer among sampled generations, without requiring the winner to exceed 50% of votes. The distinction from *“plurality voting”* is rarely enforced and may confuse the reader.

[1]. Wang, Xuezhi, et al. "Self-consistency improves chain of thought reasoning in language models." *arXiv preprint arXiv:2203.11171* (2022).

[2]. Brown, Bradley, et al. "Large language monkeys: Scaling inference compute with repeated sampling." *arXiv preprint arXiv:2407.21787* (2024).

[3]. Du, Weihua, Yiming Yang, and Sean Welleck. "Optimizing temperature for language models with multi-sample inference." *arXiv preprint arXiv:2502.05234* (2025).

**Questions:**

- See Weaknesses
- For weakness 1, could the authors provide some theoretical proof about why $5-10$ samples are enough?
- Can the methodology be applied to other test-time voting methods, like weighted majority voting and best-of-N?
- Even if the proposed estimator can accurately predict the accuracy curve of majority voting as a function of the number of samples $M$, is there any principled way to determine a *suitable* or *optimal* number of samples in practice? Without such guidance, it is unclear how this estimator informs real-world inference-time decisions.

---

### Official Review · Reviewer_rj8o · 2025-10-26

**Soundness:** 1
**Presentation:** 3
**Contribution:** 1
**Rating:** 2
**Confidence:** 4

**Summary:**

This paper aims at improving computational efficiency of majority-voting techniques used to improve LLM accuracy. The main contribution is to compute a first estimate of the LLM's output distribution and then to simulate majority-voting without an LLM.

**Strengths:**

The paper is well-written and the plots are readable.

**Weaknesses:**

The contributions of this paper are very limited.

The paper argues that if we know a good estimate of the categorical distribution $p(x)$ over output sequences of an LLM then we can speed up the majority-vote sampling process by sampling from a categorical distribution instead of the LLM. I think one could even directly compute the expected majority vote instead of simulating sampling M times from a categorical distribution (as in 4.1), provided we have a good-enough $p(x)$. Either way, the challenge is rather to estimate $p(x)$ in the first place (from my perspective).

In addition to the weak contribution, the experimental analysis does not clearly demonstrate that this simple approach is significantly better than majority-voting approaches in the literature. The current empirical evidence is very limited since (1) the approach is evaluated on just a single dataset, and (2) a detailed comparison to baselines is missing:

First, the paper argues that the computational effort is too high to extend the analysis to more datasets. The problem of the current evaluation is, however, rather that it remains unclear if this dataset is reasonable and sufficient to study the problem. Second, (and more important), it might be the case that a small number of samples is also sufficient for the baselines to obtain a good performance estimate. The paper is missing a detailed comparison with computational effort / FLOPS directly compared against method performance in estimating accuracy. In particular, the paper is missing a comparison of a small number of samples (KM) in previous approaches against the small number of samples used for this approach (G). This baseline comparison would be critical to correctly assess the contribution of this paper.

Overall, this paper does not have significant contributions, insights and empirical evidence to recommend it for acceptance.

**Questions:**

What does "ground truth" refer to exactly in Figure 2?

---

### Official Review · Reviewer_J8aT · 2025-10-29

**Soundness:** 2
**Presentation:** 2
**Contribution:** 2
**Rating:** 4
**Confidence:** 4

**Summary:**

This paper proposes a method to estimate the accuracy of the self-consistency method during "test-time scaling" (i.e., how the ensemble accuracy changes with respect to the number of samples).

**Strengths:**

1) The research problem addressed in this paper—i.e., the accuracy estimation for self-consistency method—is meaningful.
2) The introduction section is clear and easy to follow.

**Weaknesses:**

1) At the end of the paper, in Appendix D, the authors mention that, "we used a LLM, GPT-4o, to assist in retrieval and discovery of related works".
Based on this, the related work section provided in the paper is too brief, and the description of the most relevant literature—i.e., the literature on "estimating the accuracy of self-consistency method during scaling"—is lacking. The authors note in the related work section that this research problem has been proposed by prior works, but they do not provide an introduction to the methodologies proposed by the prior works.
2) In the experiments, the authors only use one dataset for evaluation. It is recommended that the authors use more datasets and cover different types of tasks, rather than focusing solely on math.

Some other concerns:
It is recommended that the authors add references the first time "test-time voting (TTV)" is mentioned in the Introduction. Additionally, it seems that this term is not frequently used by researchers in the field of test-time scaling.

**Questions:**

None.

---

### Author Response · Authors · 2025-12-04
**Major Points**

We sincerely thank the reviewers for their constructive feedback.  To support the AC with review, we provide a single unified rebuttal wherein we enumerate the most significant reviewer feedback (and which reviewer(s) provided it), followed by our responses.  Our responses focus on the most crucial high-level details (e.g., quantitative evidence and improvements), but we refer the AC to specific locations in the revised manuscript where we include full details.

- **Not enough models, datasets, or empirical evidence (fSjZ, Rj8o, J8aT).** To address the reviewers’ concern regarding empirical scope, we have substantially expanded our evaluation. The revised manuscript now includes five benchmark datasets (MATH, GSM8K, MMLU, GPQA, and CommonSenseQA) and a total of four models, providing broader coverage across reasoning, factual knowledge, and commonsense domains. We have also increased the number of samples per question to 150 for each model–dataset pair, ensuring statistically more reliable and representative results. Because we were unable to evaluate the additional datasets with GPT4ominin we have removed it from our results so that we provide a consistent analysis across all datasets for each model. We note that despite it's removal our take away points do not change.

- **Compute calculation (fSjZ, Rj8o, Ccxh)**. We appreciate the reviewer’s feedback regarding our compute calculations and have clarified their presentation in the revised manuscript. We now explicitly show that our approach achieves a one–order-of-magnitude reduction in total compute per question and an over eight–order-of-magnitude reduction in Post-G compute requirements. To improve clarity, we provide a table summarizing compute costs alongside the Mean Absolute Error for each model–method pair, and include an appendix detailing the explicit total and Post-G FLOP calculations for each voting method (empirical, monte carlo, and analytical).

- **Experimental Design & Ground Truth (fSjZ, Rj8o)**. There was some confusion about how our ground truth was obtained, so we have clarified this in the paper, and we provide a detailed explanation here.
  - Our methodology and evaluation are grounded in a key observation: a LLM can be understood as a multinomial output generator. Accordingly, the objective of our evaluation is to assess whether the proposed estimators can accurately and efficiently recover a multinomial distribution that approximates the one implicitly induced by the LLM.
To render this evaluation computationally feasible, we treat a real LLM as a multinomial generator and collect a fixed number of responses (150 total) per question. These samples are used to construct an empirical multinomial distribution over the candidate answers to a question. As illustrated in Figure 2 of the updated work, this empirical distribution then serves as a synthetic LLM, allowing a computationally-controlled comparison of our proposed estimators against baseline methods from prior work.
  - Ground Truth.  For each question and ensemble size *M*, we obtain our ground truth ground-truth accuracy by drawing *K=1,000* samples of size *M* from the synthetic LLM and averaging the resulting accuracies.

- **Simple Calculation & Baseline Comparisons.**
  - **Simple Calculation (rj8o)**. We agree that directly computing the expected majority-vote outcome would be preferable. However, to our knowledge, no closed-form method for majority-vote computation exists in the current literature. We conducted an extensive review across probability theory, statistics, computer science, and political science sources, and found only analogous formulations in probabilistic reliability theory [1]. These works define “majority” differently from its use in the LLM and ML context—requiring a response candidate to exceed 50% frequency rather than simply occur more often than all others. **This gap motivated the development of the novel closed-form solution presented in our paper. Which we believe is the best way to get a direct computation of the expected majority vote.**
[1] Kishor S. Trivedi. Probability and Statistics with Reliability, Queuing, and Computer Science Applications. John Wiley & Sons, Inc., 2016.
  - **Baseline Comparisons (Rj8o, J8aT).** We note that the only established approach to evaluating majority voting in the literature relies on empirical trials; we found no prior use of estimation or approximation methods. While this point was already mentioned in our related work, we have now made it more explicit in the revised text. For improved clarity, we replaced Figure 2 with a table reporting the Mean Absolute Error between our proposed methods and this standard baseline. As shown, our methods remain low-error and precise, further supported by the expanded set of models and datasets described above.

---

> ### Author Response · Authors · 2025-12-04
> **Additional Significant Points**
>
> - **Fundamentally Flawed Assumption (Ccxh)**. The reviewer’s comment appears to stem from a misunderstanding of our formulation. Equation (10) does not rely on the assumption described in the review. In fact, the approach the reviewer suggests aligns with the method we introduce in the paper (see lines 245–250 and 257–268). We have clarified this connection in the revised manuscript to prevent further confusion.
>   - We have also added clarification in the results section to explain why covariance need not be explicitly modeled in this context. For concentrated probability mass functions (PMFs)—as shown in our original Figure 3—correlations between candidates are negligible, since the PMF shape itself encodes the dependence strength.
>   - Empirically, our results table demonstrates that this assumption holds across a broad range of settings: it yields low approximation error across four models and five datasets. These results confirm that the proposed method is both valid and robust, and does not exhibit the flaw suggested in the review.
>
> - **Justification for a Small Sample Size (fSjZ).** Our original Figure 3 shows that, on average, the majority of the probability mass function (PMF) for a question is concentrated in 5–10 response candidates, indicating that G=5–10 is sufficient to approximate the majority-vote outcome. This finding is further supported in our new results table, which demonstrates consistently low approximation error across multiple models and datasets at these sample sizes. Finally, the temperature and difficulty experiments (Figures 3 and 4) were specifically designed to test the limits of our assumptions; we found that under challenging questions (e.g., where p^{*}<=0.25, G=5–10) and non-standard operating conditions (temperature=2.0 or 8.0) G=5 or 10 remained sufficient. The higher temperature settings were intentionally used to probe failure modes and confirm robustness. These points are made clearer and highlighted in our revised version.

---

> > ### Author Response · Authors · 2025-12-04
> > **Minor Points**
> >
> > - **The monte carlo method is just the parametric bootstrap.** We acknowledge the similarity between our Monte Carlo approach and the parametric bootstrap. However, to our knowledge, neither bootstrap nor Monte Carlo methods have been previously applied to majority-vote estimation in the LLM context. The novelty of our contribution lies in adapting this well-understood technique to a new problem domain, where it yields substantial practical benefits—including an order-of-magnitude reduction in compute cost. In combination with our other results, this provides a valuable new baseline for approximation methods, which, as we note, are largely absent from the existing literature.
> >
> > - **More Literature Review (J8aT).** We conducted an extensive literature review as part of the original submission, combining manual search with LLM-assisted retrieval to ensure comprehensive coverage. This process identified nine high-impact papers from major LLM developers (OpenAI, Google, Anthropic, and others) that employ majority-vote methods consistent with ours. Given this breadth and the absence of specific missing references, we believe the current related work section sufficiently represents the relevant literature.
> >
> > - **Formatting & Naming Conventions (J8aT, fSjZ, Ccxh).** All noted issues have been addressed, and the manuscript has been revised for uniform terminology, consistent variable naming, and improved readability throughout.

---

### Note · Authors · 2025-12-04

**Comment:**

We are unable to upload our revised manuscript.

**Withdrawal Confirmation:**

I have read and agree with the venue's withdrawal policy on behalf of myself and my co-authors.